# Environmental Signals Act as a Driving Force for Metabolic and Defense Responses in the Antarctic Plant *Colobanthus quitensis*

**DOI:** 10.3390/plants11223176

**Published:** 2022-11-21

**Authors:** Laura Bertini, Silvia Proietti, Benedetta Fongaro, Aleš Holfeld, Paola Picotti, Gaia Salvatore Falconieri, Elisabetta Bizzarri, Gloria Capaldi, Patrizia Polverino de Laureto, Carla Caruso

**Affiliations:** 1Department of Ecological and Biological Sciences, University of Tuscia, 01100 Viterbo, Italy; 2Department of Pharmaceutical and Pharmacological Sciences, University of Padova, 35100 Padova, Italy; 3Institute of Molecular Systems Biology, Department of Biology, ETH Zurich, 8093 Zurich, Switzerland

**Keywords:** *Colobanthus quitensis*, differential proteomic analysis, environmental signals, enzymatic activity, gene expression analysis, MS/MS analysis, response to stress

## Abstract

During evolution, plants have faced countless stresses of both biotic and abiotic nature developing very effective mechanisms able to perceive and counteract adverse signals. The biggest challenge is the ability to fine-tune the trade-off between plant growth and stress resistance. The Antarctic plant *Colobanthus quitensis* has managed to survive the adverse environmental conditions of the white continent and can be considered a wonderful example of adaptation to prohibitive conditions for millions of other plant species. Due to the progressive environmental change that the Antarctic Peninsula has undergone over time, a more comprehensive overview of the metabolic features of *C. quitensis* becomes particularly interesting to assess its ability to respond to environmental stresses. To this end, a differential proteomic approach was used to study the response of *C. quitensis* to different environmental cues. Many differentially expressed proteins were identified highlighting the rewiring of metabolic pathways as well as defense responses. Finally, a different modulation of oxidative stress response between different environmental sites was observed. The data collected in this paper add knowledge on the impact of environmental stimuli on plant metabolism and stress response by providing useful information on the trade-off between plant growth and defense mechanisms.

## 1. Introduction

Antarctica was the last continent to be discovered, likely due to its harsh environment and geographical isolation from other regions of the Earth. Climatic and environmental conditions are so severe that they do not allow the development of numerous species as in other regions of our planet. Indeed, this holds true especially for vascular plants that grow only in the Antarctic Peninsula, which is characterized by milder conditions. In fact, its temperatures are warmer, exceeding 0 °C during the Antarctic summer and rarely dropping below −10 °C during the Antarctic winter [1]. Nonetheless, sub-zero temperatures are also characteristic of the austral summer during the night, particularly on King George Island (South Shetland) [2,3]. Although these conditions are milder than those of the Antarctic continent, Maritime Antarctica still represents an extreme ecosystem where the inhabiting organisms experience low temperatures, restricted availability of water and nutrients, high radiation, and wind abrasion [4,5,6,7].

The Antarctic Peninsula hosts two species of endemic flowering plants: the Antarctic hairgrass *Deschampsia antarctica* E. Desv. (Poaceae) and the Antarctic pearlwort *Colobanthus quitensis* (Kunth) Bartl. (Caryophyllaceae) [8]. Furthermore, two more species were introduced accidentally in this region, both belonging to the Poaceae family, i.e., *Poa pratensis* L. and *Poa annua* L. [9,10]. In addition, one more non-native plant belonging to the Juncaceae family has been identified in association with the endemic Antarctic plants, i.e., *Juncus bufonius* L. [11]. Apart from the newly introduced species, only *D. antarctica* and *C. quitensis* have been able to naturally colonize a vast part of Maritime Antarctica down to ca. 68° S, spreading to the west coast of the Antarctic Peninsula and its associated islands [12]. Indeed, *C. quitensis* has a wider area of colonization, also extending along the Andes to Ecuador, with a site in Mexico [13]. 

These plant species have developed a moderate to perfect adaptation to cold and frost and have also been experiencing the effect of rising temperatures over the last decades. Both species are able to acclimate to the cold by modulating their LT50 (lethal temperature at 50%), i.e., the temperature at which 50% of the leaf tissue dies due to freezing, and are therefore considered freezing-tolerant species [2,14]. Along with the mechanisms of freezing resistance, other biochemical, physiological, and morphological adaptations occurred during evolution, allowing their survival and spread in the harsh Antarctic environment. Anatomical and ultrastructural modifications have been reported for *D. antarctica* which has several xerophytic characteristics, such as small and thick leaves, high stomata density per area, thick cuticle, and high morphological plasticity of organs and organelles [15]. *C. quitensis* has linear and sessile leaves also showing typically xeric characteristics, such as high thickness, higher density of diacytic stomata found on both leaf surfaces (amphistomatic leaf), and the presence of the bundle sheath that minimizes apoplastic water movement toward mesophyll particularly dense and poor in cell wall fibers [8]. Furthermore, *C. quitensis* has a cushion conformation called pearlwort that allows reduced exposure to abiotic stresses such as strong winds and poor water availability compared to *D. antarctica*. In addition, both plants developed many physiological adaptations such as the ability to maintain a positive photosynthetic rate near 0 °C, resistance to photoinhibitory conditions, and tolerance to water stress [8]. Furthermore, it has also been demonstrated that *C. quitensis* populations show anatomical and physiological adaptation along a latitudinal gradient, and individuals inhabiting cold zones at high latitudes increase their ecophysiological performance under simulated global warming conditions more than northernmost populations [16]. These plants have been recognized over the past years as bioindicators of climate change. Indeed, it has been hypothesized that their expansion and diffusion could be mainly triggered by summer air warming [17,18]. Beyond temperature change, extremophilic plants also have to cope with other harsh environmental conditions that can act as a trigger for the activation of defense mechanisms against (a)biotic stress or adaptation strategies. Among the latter, the role of endophytic microorganisms is emerging as one of the many strategies to deal with extreme environmental conditions, although the peculiar traits are starting to be deepened [19,20,21].

In recent years, we performed the de novo transcriptome assembly of *C. quitensis* plants grown in a low-temperature natural habitat compared to plants grown for one year inside open-top chambers (OTCs), which determine an increase of about 4 °C at midday [22]. In addition, we shed some light on the proteome remodeling of *C. quitensis* grown under the same experimental conditions using a differential proteomic approach [23]. Overall, these results reveal that *C. quitensis* plants grown at warmer temperatures display a high rate of photorespiration which likely acts as a protective mechanism against photooxidative damage, ROS production, and lipid peroxidation [23]. These results are in agreement with those reported by Cho et al. [24] who compared the transcriptome of *C. quitensis* plants grown in the natural cold habitat versus those grown under milder growth conditions in the laboratory.

Due to the progressive environmental change to which the Antarctic Peninsula is subjected over time, a more comprehensive overview of the metabolic characteristics of local plants becomes particularly interesting for evaluating their ability to deal with environmental stresses. This work aimed to investigate the *C. quitensis* proteome rewiring triggered by environmental cues by using an integrated differential proteomic approach. To this end, plants from three different sites were analyzed. The sites differ in several environmental conditions near the Antarctic Polish base Henrik Arctowski (King George Island, South Shetland). Among them are the distance from the coastline, altitude, air temperature, wind speed, and soil composition. The data collected in this work, combined with the information available in the literature, add knowledge on the impact of environmental stimuli on plant metabolism and stress response by providing useful information on the trade-off between plant growth and defense mechanisms.

## 2. Results and Discussion

This work aimed to study the metabolic reprogramming of *C. quitensis* triggered by environmental cues using a combination of high-throughput proteomic profiling techniques and bioinformatics tools to obtain a large-scale study of hundreds of proteins expressed under specific conditions within the plant. In particular, we drew our attention to the study of *C. quitensis* growing in three different sites (Site 1, Site 2, Site 3) (Appendix A) near the Polish Antarctic base Henryk Arctowski (King George Island, South Shetland), whose characteristics are summarized in Table 1.

The different positions of the three sites correlate with differences in plant growth conditions. In particular, the distance from the coastline differently affects the soil salinity. Indeed, Site 1 is located near the beach, about 90 m from the coastline, thus receiving sea spray, and is influenced by guano and feces of birds and mammals that live there. On the other hand, Sites 2 and 3 are located at progressively higher altitudes (Site 3) that expose the plants to colder winds than at sea level. Furthermore, the three sites show different textures of soils which are associated with different degrees of permeability and, consequently, with different water availability for plants.

### 2.1. Determination of the Differentially Expressed Protein by a Proteomic Approach

The response of *C. quitensis* to the different environmental cues was determined by analyzing changes in protein expression by using a comparative mass spectrometry (MS)-based proteomics approach (Figure 1). *C. quitensis* leaves were collected in Sites 1, 2, and 3 described above as highlighted in Figure 1, during the summer season of 2020, and the proteins were extracted by a procedure optimized to obtain a good yield [25].

The complete mass spectrometry data for each data set are reported in Appendix A. A total of 5103 proteins (of which 2872 proteins with at least two peptides) were identified by searching against the in-house *Colobanthus quitensis* protein database [23]. The variations of protein expression profiles were evaluated using a data-independent acquisition (DIA) quantitative proteomics approach that allows quantifying proteins based on the relative intensities of fragment ions. Finally, the data were normalized across the different conditions, and a t-test from the *limma* R package was applied to assess statistical significance, setting the significance threshold (*p*-value) at *p* = 0.05. 

Using this approach, we identified 412 differentially expressed proteins (DEPs) by comparing the sampling sites in pairs, obtaining three different data sets (i.e., S2 vs. S1, S3 vs. S1, and S3 vs. S2). Fold change (FC) was calculated as the ratio SpCsS2/SpCsS1, SpCsS3/SpCsS1, and SpCsS3/SpCsS2. To identify statistically significant DEPs in each data set, a |log_2_FC| >0.58 and a *p*-value <0.05 were considered. Detailed results are shown in Figure 2.

In this work, the pBLAST tool was used to identify the *Arabidopsis thaliana* orthologues of the differentially expressed proteins disclosed by the proteomic approach (https://blast.ncbi.nlm.nih.gov/Blast.cgi?PAGE=Proteins, accessed on 4 April 2022). Choosing the Arabidopsis proteins with the highest percentage of identity with our data sets, the highest sequence coverage, and the lowest E-value, the Arabidopsis orthologous proteins and their codes, known as TAIR (The Arabidopsis Information Resource) codes, were obtained. These codes, which represent the unique identifier of the *Arabidopsis thaliana* genes present in the TAIR database (https://www.arabidopsis.org/, TAIR 10 release, accessed 11 April 2022), were used for all bioinformatics analyses aimed at characterizing the identified *C. quitensis* DEPs in all data sets. Protein descriptions of the up- and downregulated DEPs identified in all data sets are reported in Appendix A along with log_2_FC, *p*-values, TAIR codes, and TAIR protein descriptions. It is worth mentioning that only in a few cases, different isoforms share the same TAIR code with the same protein description. Consequently, the number of DEPs identified by the TAIR codes in each data set is slightly lower than that reported in Appendix A.

### 2.2. Gene Ontology Analysis

In order to deepen our knowledge of the biological relevance of the identified DEPs, we performed a Gene Ontology (GO) analysis. GO enrichment analysis was performed using the freely available ShinyGO software version 0.76, created by Xijin Ge and Jianli Qi of South Dakota State University (Brookings, South Dakota, US), (http://bioinformatics.sdstate.edu/go/, accessed on 26 April 2022) [26] using the TAIR code of Arabidopsis orthologous proteins as input. The three GO domains “Biological Process”, “Molecular Function”, and “Cellular Component” were considered in the analysis. Nevertheless, to better understand *C. quitensis* response to different environmental conditions, a more accurate investigation was carried out within the Biological Process domain. 

#### 2.2.1. Data set S2 vs. S1

Figure 3 shows a graphical representation of the significantly enriched GO terms obtained with the ShinyGO tool. In particular, 81 and 45 proteins were found to be upregulated and downregulated in this data set, respectively.

As for the upregulated DEPs, many proteins were found to be responsive to abiotic stress as “response to cadmium/metal ion” or, more in general, “response to inorganic substances”. The presence of heavy metal pollution in Antarctica has been reported in the last few years, although it is commonly deemed a pristine continent. In particular, the contamination of the western shore of the Antarctic Peninsula by heavy metals has been recently highlighted and suggested as a proxy for climate change in Antarctica [27]. Indeed, there is a great concern about the impact of anthropogenic activities on the environment, especially in the vicinity of the research stations [28], where human activities are the major cause of pollution. Nevertheless, the tectonic origin of inorganic compounds continuously washed away by the melting of regional glaciers cannot be ruled out [27]. Interestingly, the cadmium ion has been shown to trigger the overproduction of reactive oxygen species (ROS), including H_2_O_2_ [29], and to inactivate several enzymes by binding with sulfhydryl groups of cysteine and by replacing iron ion from various proteins [30]. More in general, cadmium ion can be regarded as an activator of antioxidant responses, which, conversely, make plants capable of counteracting oxidative stress [31]. Indeed, among the upregulated DEPs in this data set (Appendix A), we can highlight some “ROS scavenger” proteins, known to improve plant tolerance to oxidative stress. Namely, manganese superoxide dismutase 1 (MSD1, AT3G10920), which catalyzes the conversion of the superoxide radical to O_2_ and H_2_O_2_, peroxidase 12 (PER12, AT1G71695), involved in the response to oxidative stress, and monodehydroascorbate reductase (MDAR1, AT3G52880), involved in the ascorbate–glutathione cycle, which removes toxic H2O2 (Appendix A). All enzymes are components of the cellular antioxidant defense system and are also able to respond to metal ions such as cadmium, as suggested in the literature. 

Within the upregulated DEPs, many GO terms are also related to the response to biotic stimulus (i.e., “response to bacterium”, “response to other organisms”, or “interspecies interaction between organisms”), highlighting the ability of plants growing in Site 2 to activate plant immunity and defense response as well (Figure 3). In particular, CDC5 (AT1G09770) is noteworthy for its countless features. CDC5 protein is a conserved protein in animals, plants, and fungi [32]. It was first isolated from *Schizosaccharomyces pombe* as a cell cycle regulator and is considered a putative transcription factor, belonging to the MYB-related protein family [32,33]. Furthermore, CDC5 has been shown to act as a component of the spliceosome to participate in mRNA splicing in humans and yeast [34,35]. In Arabidopsis, CDC5 binds DNA and is required for development and immunity to bacterial infection [36,37]. Besides the transcription factor CDC5, proteins involved in plant defense against other organisms, such as several pathogenesis-related proteins (PRs), have been highlighted. PRs are widely recognized to play a key role in plant defense mechanisms activated by both biotic and abiotic stresses [38]. These proteins not only accumulate locally at the infection site but are also systemically induced concomitantly with the development of systemic acquired resistance (SAR) [39]. In our study, several PR proteins were disclosed as glucanase BG3 (PR2, AT3G57240), chitinase EP3 (PR3, AT3G54420), plant defensin PDF2.1 (PR12, AT2G02120), and lipid transfer protein LTP3 (PR14, AT5G59320), which were found to be overexpressed in plants of Site 2 (Appendix A). Furthermore, polygalacturonase-inhibiting protein 1 (PGIP1, AT5G06860) and azelaic acid-induced protein 1 (AZI1, AT4G12470) have been recognized within this data set (Appendix A), both involved in response to fungal infection and reported as SAR activators [40,41]. 

Finally, several upregulated proteins were disclosed within the “carbohydrate metabolic process” term (Figure 3), such as hexokinase1 (HK1, AT4G29130), involved in glycolysis, chloroplastic fructose-bisphosphate aldolase (FBA, AT2G01140), which takes part in the regeneration phase of the Calvin–Benson cycle [42], and phosphoenolpyruvate carboxylase (PEPC1, AT1G53310), involved in both carbon and nitrogen metabolism (Appendix A) [43]. Furthermore, the presence of pyruvate dehydrogenase E1 alpha subunit (PDH, AT1G01090), involved in the formation of acetylCoA, NADH-ubiquinone oxidoreductase (AT3G03100), playing a role in the electron transport chain and photorespiration, and ATP synthase (ATPase, V1 complex, subunit B protein, AT1G20260), strongly suggests the activation of the cellular respiration pathway in plants growing at Site 2 (Appendix A). 

Among the downregulated DEPs in the S2 vs. S1 data set (Figure 3), the most represented terms are related to the response to protein folding and unfolding. The identification of different molecular chaperones, known to improve plant tolerance to thermal shock, deserves remarkable importance. In particular, six heat shock proteins (HSPs) have been disclosed within this group (Appendix A): HSP17.6II (AT5G12020), HSP 18.2 (AT5G59720), HSP20-like (AT1G53540), HSP70 (AT3G12580), HSP90.1 (AT5G52640), and HSP101 (AT1G74310). Some of them are also involved in response to high light intensity, response to hydrogen peroxide, and heat acclimation [44,45,46]. Interestingly, several HSPs also respond to different environmental signals that induce the accumulation of unfolded and misfolded proteins in the endoplasmic reticulum (ER), causing ER stress [47]. Cells defend themselves from ER stress by activating a signal transduction pathway, termed unfolded protein response (UPR), which enhances protein folding and/or removes unfolded/misfolded proteins from the ER [48]. Three more proteins specifically involved in the ER stress response were identified in this data set: protein DNAJ (also known as HSP40, AT2G20560, AT2G22360), binding protein 2 (BIP2, AT5G42020), and protein disulfide isomerase (PDI-LIKE 2-2, AT1G04980) (Appendix A). DNAJ belongs to a protein family containing the J domain, which interacts with HSP70 heat shock proteins regulating their ATPase activity [49,50]. BIP2 belongs to an ER molecular chaperone protein family that has been shown to assist protein folding and to act in the ER quality control mechanism that recognizes unfolded or abnormally folded proteins and sends them out of the organelle for degradation [51]. PDI-LIKE 2-2 catalyzes disulfide bond formation and acts as a molecular chaperone in assisting polypeptide folding [52]. PDI-LIKE 2-2 is also a component of the unfolded protein response that relieves ER stress and reduces programmed cell death [53]. As expected, using the freely available STRING program, it was demonstrated that all proteins involved in the ER stress response interact with each other, confirming their involvement in a network of physiological processes strictly connected to the control of *C. quitensis* protein folding (Appendix A). 

Finally, a number of DEPs responsive to saline stress were also disclosed in the S2 vs. S1 data set, such as Bcl-2-associated athanogene 4 (BAG4, AT3G51780), belonging to the co-chaperone regulator protein family [54], and aldo/keto reductase (AKR, AT1G59960), which belongs to the AKR NAD(P)H-dependent protein family [55] (Appendix A).

Overall, the greater metabolic activity of plants of Site 2 compared to those of Site 1 may suggest that the former face less stressful environmental conditions, allowing both a higher metabolic rate and greater protection against stress of different origins. Based on these results, we can speculate that the plants of Site 2 are more resilient and better suited to counteract adverse environmental conditions.

#### 2.2.2. Data set S3 vs. S1

A graphical representation of the GO terms significantly enriched in the up- and downregulated protein data set S3 vs. S1, as obtained with the ShinyGO tool, is reported in Figure 4.

It is worth mentioning that fewer DEPs were highlighted within this data set. In particular, 50 and 28 DEPs were disclosed within the up- and downregulated DEPs, respectively. Proteins involved in the response to abiotic stimuli such as cold, water deprivation, and abscisic acid (ABA) were identified within the upregulated DEPs. Among them are cold-regulated 47 (COR47, AT1G20440) and RAB18 (AT5G66400) (Appendix A), both belonging to the dehydrin protein family and showing sequence homology with the late embryogenesis abundant (LEA) proteins [56]. Arabidopsis plants overexpressing both COR47 and RAB18 were found to be more cold-tolerant [57]. Furthermore, glycine-rich RNA-binding protein 2 (GR-RBP2, AT4G13850) and RNA-binding ribosomal protein S1-like (SRRP1, AT3G23700), both involved in response to cold, osmotic stress, and ABA, were also disclosed [58,59] (Appendix A). In addition, two proteins belonging to the family of lipid transfer proteins (LTPs), involved in the transfer of phospholipids to the membrane in response to environmental stresses, were identified (LTP2, AT2G38530; LTP3, AT5G59320) (Appendix A) [60,61].

Besides abiotic stress response, we wondered whether DEPs involved in primary metabolism were present in this data set to shed light on the impact of different environmental conditions on plant growth and development. To this end, the freely available KEGG pathway tool was used [62] highlighting four proteins grouped under the general term “metabolic pathways”, namely, the granule-bound starch synthase 1 (GBSS1, AT1G32900), belonging to the UDP-Glycosyltransferase superfamily, the dUTP-pyrophosphatase-like 1 (DUT1, AT3G46940), ATPase (delta/epsilon subunit, AT5G47030), and nitrite reductase 1 (NIR1, AT2G15620 (Appendix A). We focused our attention on NIR1 as this protein was found to be overexpressed also in the data set S3 vs. S2, suggesting that nitrogen metabolism is more active in plants from Site 3 than in those growing at the other sites. Nitrate assimilation is essential for plant growth as it is the predominant N source in soils. Its reductive assimilation requires the activity of soluble cytosolic NADH-nitrate reductases (NRs) and plastid stroma ferredoxin-nitrite reductases (NIR) allowing the reduction of nitrate to nitrite and then to ammonium [63,64]. It is worth mentioning that under hypoxic conditions, nitrite can be reduced to nitric oxide (NO) in the mitochondria [65] or in the cytoplasm, through an NR-catalyzed side-reaction [66]. The overexpression of NIR1 in plants growing in Site 3 may cause an imbalance of NO homeostasis leading to a decrease in this signal molecule in favor of the production of ammonium. This event has a great impact on plant fitness as NO has emerged for many years as a powerful regulator of plant growth and development, as well as a key signal molecule involved in environmental stress and pathogen responses [67]. Therefore, the increased expression of NIR1 in plants growing at Site 3 could lead to a decreased amount of NO which in turn could have a negative impact on plant resilience and defense compared to plants growing at Sites 1 and 2.

As for the downregulated DEPs, some of them were categorized within the “Innate immune response” term and related domains, such as the plant immune receptor RPM1 (Resistance to *Pseudomonas syringae* pv. Maculicola 1, AT3G07040) (Appendix A) known to recognize pathogen-released effectors to activate effector-triggered immunity (ETI) in *Arabidopsis thaliana* [68]. It is noteworthy that RPM1 was also found downregulated in the S3 vs. S2 data set, as reported in the next section. Moreover, in this domain, we also highlighted the transcription factor CDC5, which was also found upregulated in the S2 vs. S1 data set and downregulated in the S3 vs. S2 data set, as discussed in the next section (Appendix A). In addition, several proteins implicated in the translocation of photosynthetic proteins from the cytosol to the chloroplast were disclosed within the term “Protein import/transport”. Chloroplasts use two protein translocation systems, one through the outer envelope (TOC) and the other through the inner envelope (TIC) [69]. We found that the TIC40 (AT5G16620) protein, besides its function in protein import to chloroplasts, is also involved in processes leading to thylakoid biogenesis [70]. Furthermore, the outer plastid envelope protein OEP16 (AT2G28900), involved in the plastid import of NADPH:protochlorophyllide oxidoreductase A (PORA) [71], and the translocase of outer mitochondrial membrane 20 kDa subunit 3 (TOM20-3, AT3G27080) [72] were also discovered. Finally, the presence of ROS scavenger enzymes, such as a putative peroxidase localized in the endomembrane system (PER 27, AT5G15180) and a microsomal ascorbate peroxidase 3 (APX3, AT4G35000), both involved in the response to oxidative stress, is noteworthy (Appendix A).

Altogether, our results highlight that Site 3 plants trigger responses to cold, water deprivation, and ABA while appearing more vulnerable to other (a)biotic or oxidative stresses. This can be explained by the greater exposure of plants in Site 3 to lower temperatures, freezing winds, and lower water availability due to the nature of the soil. In addition, the increased expression of NIR1 could lead to a decreasing amount of NO, which in turn could have a negative impact on both plant fitness and stress tolerance when comparing plants growing in Site 3 versus Site 1. Finally, we could hypothesize that the plants of Site 3 are less efficient in the protein translocation systems, affecting both photosynthetic efficiency and energy production.

#### 2.2.3. Data set S3 vs. S2

The GO analysis carried out with the ShinyGO tool highlighted 55 upregulated and 105 downregulated DEPs within the “Biological process” domain (Figure 5).

Several DEPs were disclosed that are grouped within the response to stress as temperature (heat and cold), chemical (inorganic substances), or abiotic stress. Among them are germin3 (GER3, AT5G20630) and glycine-rich RNA-binding protein 2 (GR-RBP2, AT4G13850), involved in the response to cold [73,74], and the thiazole biosynthetic enzyme (THI1/4, AT5G54770) with the double function of biosynthesis of thiamine and mitochondrial DNA damage tolerance, involved in the ABA response [75]. Moreover, some PR proteins known to respond to abiotic stress [35] were disclosed within this data set such as thaumatin-like (AT2G28790), PDF 2.3 (AT2G02130), and lipid transfer protein 2 (LTP2, AT2G38530) belonging to the PR5, PR12, and PR14 protein family, respectively. In addition, several proteins involved in the response to protein folding and refolding as well as in the endoplasmic reticulum (ER) stress response were disclosed. Many of them were identified and extensively discussed in the downregulated S2 vs. S1 data set. Among them are many HSPs such as HSP17.6II (AT5G12020), HSP18.2 (AT5G59720), HSP21 (AT4G27670), HSP70 (AT3G12580), and ATHSP101 (AT1G74310) as well as DNAJ (AT2G20560, AT2G22360) and BIP2 (AT5G42020) (Appendix A).

With regard to downregulated DEPs, the identified GO terms can be grouped into two general domains, i.e., response to (a)biotic stress and energy metabolism. As for the response to the stress terms, we found proteins involved in the response to biotic stress, in particular response to fungal and bacterial infections, such as beta-1,3-glucanase 3 (BG3, AT3G57240), glutamate-cysteine ligase (GSH1, AT4G23100), ethylene-forming enzyme (EFE, AT1G05010), and constitutive disease resistance 1 (CDR1, AT5G33340) (Appendix A). It is noteworthy that we highlighted the presence of some proteins already disclosed in other data sets involved in response to fungal infection, SAR, and hypersensitive response. In particular, AZI1 (AT4G12470), RPM1 (AT3G07040), and PGIP1 (AT5G06860) are all upregulated in the S2 vs. S1 data set, and CDC5 (AT1G09770) is upregulated in the S2 vs. S1 and downregulated in the S3 vs. S1 data sets (Appendix A). Taken together, the latest results highlight that the proteins involved in defense mechanisms against biotic agents are more expressed in plants growing in Site 2 where they could exert a greater protective action than in plants growing in the other sites. Moreover, DEPs involved in metal and cadmium ion response were disclosed, such as peptidase M1 family protein (AT1G63770), caffeoyl-CoA 3-O-methyltransferase (AT4G34050), and mitochondrial lipoamide dehydrogenase 1 (mtLPD1, AT1G48030), induced also upon light stimulus (Appendix A). It is noteworthy that, in this data set, we highlighted once again APX3 (AT4G35000) and PER 27 (AT5G15180) (Appendix A), which were found to be downregulated in Site 3 plants compared to Site 1 as well. This result reinforces previous findings indicating that plants growing in Site 3 are more subjected to oxidative stress.

Finally, several downregulated DEPs fall in processes linked to primary metabolism and, to a lesser extent, to secondary metabolism. For instance, we can highlight several proteins involved in glucose catabolism, such as glucose-6-phosphate isomerase (PGI1, AT4G24620), phosphofructokinase (PFK, AT1G76550), glucose-6-phosphate dehydrogenase 6 (G6PD6, AT5G40760), 6-phosphogluconate dehydrogenase (6PGD, AT4G29120), transketolase-2 (TKL-2, AT2G45290), mitochondrial pyruvate carrier 3 (MPC3, AT4G22310), mitochondrial dihydrolipoyl dehydrogenase 1 (mtLPD1, AT1G48030), citrate synthase 4 (CSY4, AT2G44350), aconitase 3 (ACO3, AT2G05710), and fumarase 1 (FUM1, AT2G47510) (Appendix A). Downregulation of these proteins in the S3 vs. S2 data set implies their overexpression in plants growing in Site 2, indicating that metabolic pathways linked to energy production and growth are active in the latter plants. The same holds true for amino acid metabolism since many proteins involved in this pathway were detected as downregulated in the S3 vs. S2 data set. Among them are lysyl-tRNA synthetase 1 (KRS-1, AT3G11710) and a protease belonging to the aspartyl family protein (CDR1, AT5G33340). Furthermore, proteins involved in secondary metabolic processes were also found, such as glutathione-S-transferase TAU (GSTU19, AT1G78380), playing a positive role in drought, salt, and oxidative stress tolerance [76], and hydroxycinnamoyl-CoA shikimate/quinate transferase (HCT, AT5G48930), involved in auxin homeostasis, lignin biosynthetic process, positive regulation of flavonoid biosynthesis, and phenylpropanoid pathways [77].

Definitely, numerous similarities can be highlighted between the S3 vs. S2 and S2 vs. S1 data sets, leading to the general conclusion that plants at Site 2 may be able to defend themselves better than those growing in the other sites. Indeed, they seem to be suffering less from ER stress, are more protected from oxidative damage and (a)biotic stresses, and have a more active primary and secondary metabolism.

### 2.3. Antioxidant Enzyme Activity Assays

In order to experimentally validate the proteomic results related to the plant responses to oxidative stress, we compared the activity of some antioxidant enzymes in all samples. As shown in Figure 6, the activities of glutathione S-transferase (GST), guaiacol peroxidase (POD), and superoxide dismutase (SOD) were all significantly higher in samples of Site 2 plants with respect to the other samples (see Section 3.5 for SOD results interpretation). 

These results are in agreement with the differential proteomic data, suggesting that plants growing in Site 2 are able to activate strong antioxidant defenses to protect themselves from oxidative stress. Nevertheless, it is worth noting that the catalase (CAT) activity was found to be significantly overexpressed in Site 3 plants, therefore apparently in contrast with what has just been stated. As mentioned before, the three sampling sites differ in numerous characteristics (Table 1), including the composition of the soils. In particular, Site 3 is characterized by a higher percentage of clay (16%) and silt (27%) and a lower percentage of sand (57%), compared to the other sites. This makes the soil of Site 3 very compact, especially in conditions of scarce water availability, as occurs in Antarctica. Consequently, Site 3 soil is characterized by fewer infiltrations of air, which deprive the roots of oxygen and impair their development. Indeed, it has been reported that plants growing in Site 3 have to constantly face stresses that affect their growth and spread [2]. Besides this, it has been reported that under drought conditions, ABA prevents H_2_O_2_ accumulation through the induction of catalases [78]. Based on our results, we can hypothesize that the increased CAT activity found at Site 3 could be related to soil texture resulting in a drought-like condition. The presence of several DEPs involved in the ABA response in plants growing in Site 3 further corroborates this hypothesis. 

To assess membrane damage caused by increased ROS levels, TBARS content generated by lipid hydroperoxides was measured. Lipid peroxidation is a harmful process in plants, which affects the properties of biological membranes, compromising their functions as ATP production in the mitochondria, protein synthesis in the endoplasmic reticulum (ER), and protein post-translational modification in the Golgi [79]. Furthermore, lipid peroxidation products have been found to mediate stress-induced damage in plants as well as programmed cell death (PCD) [80]. As shown in Figure 6, the TBARS content was found to be significantly higher in Site 3 compared to Sites 2 and 1 which did not show significant differences between them, confirming that the plants growing in Site 3 are more damaged than in other sites. 

In conclusion, and in agreement with the results of differential proteomics, plants growing in Site 3 appear to be less tolerant to oxidative stress due to the low activity of ROS scavenger enzymes, leading to greater oxidative damage in terms of lipid peroxidation. Conversely, plants growing in Site 2 are able to activate strong antioxidant defenses (Figure 6), as evidenced by the increased activity of scavenger enzymes, and, consequently, are less damaged by oxidative stress as shown by the TBARS assay. Overall, our results highlight that the plants growing in Site 2 have a higher ability to counteract stressful conditions and appear to be more efficient to cope with the harsh environmental conditions in which they grow, exhibiting better performance and tolerance to stress.

### 2.4. Gene Expression Analysis

In order to validate the results obtained with the differential proteomic analysis, the expression pattern of specific genes highlighted by the GO was determined by qPCR. In particular, we evidenced that metabolic pathways linked to energy production and growth are more active in plants growing in Site 2 than in the other sites. To corroborate these results, we selected hexokinase 1 (HK1) and transketolase-2 (TKL), which were both upregulated in the S2 vs. S1 data set. Furthermore, TKL was found to be downregulated in the S3 vs. S1 data set. As shown in Figure 7, both genes are overexpressed in S2 samples, confirming the proteomic data. 

Moreover, as evidenced by the differential proteomic data, plants growing in Site 3 were found to produce a higher level of NIR1 compared to the other samples. According to the literature [81], this finding could have a negative impact on the resilience and defense of plants as it could reduce NO levels, thus limiting its positive potential in promoting growth and stress tolerance. Gene expression analysis of NIR1 confirmed the proteomic data showing that it is significantly overexpressed in plants growing at Site 3 compared to Sites 2 and 1 (Figure 7), thus suggesting that Site 3 plants may suffer more from the harsh environmental conditions they face.

As previously reported, proteomic analyses also disclosed a protein involved in plant defense response and immunity, namely CDC5. This protein was found to be differentially expressed in all three data sets, and, in particular, it was found upregulated in the S2 vs. S1 data set and downregulated in both S3 vs. S1 and S3 vs. S2 data sets (Appendix A). CDC5 belongs to the MYB-type transcription factor [32,33] and plays a pivotal role in the activation and onset of the plant’s immune system against bacterial infections [36,37]. Gene expression analysis by qPCR fully confirmed the results obtained by proteomics, further supporting the hypothesis that plants of Site 2 respond more efficiently to biotic infections. In addition, the lower expression level of CDC5 in S3 plants confirms that plants growing in this site are more prone to biotic stress. 

Finally, beyond their involvement in the specific response to ER stress, molecular chaperones (HSPs) are widely recognized as proteins involved in the response to many types of stress, especially temperature variation. Our proteomic analyses disclosed several proteins falling in this protein family or related to it. In particular, we found several HSP proteins significantly upregulated in the plants of Site 1 and Site 3 compared to Site2, supporting the hypothesis that the latter is less challenged by environmental stresses. To confirm this evidence, we chose the DNAJ protein, also called HSP40, as a molecular marker [82]. As reported in Figure 7, the qPCR analysis of DNAJ confirmed its downregulation in Site 2 plants compared to those of Site 1 and even more of Site 3. This further validation sustains the reliability of the proteomics data obtained and supports the idea that the plants of Site 2 are subject to environmental conditions more favorable for growth and tolerance to stress than the other sites, thus limiting the activation of defense responses to stress. This result is in agreement with Sierra-Almeida et al. [2] who stated that Site 2 appears to be more favorable for plant growth due to well-drained soils and nutrient availability and where individual plants grow bigger [83].

## 3. Materials and Methods

### 3.1. Sample Collection

The sampling area is located near the Polish research station Henryk Arctowski (62°14′ S, 58°48′ W) overlooking Admiralty Bay, a bay on King George Island, South Shetland Archipelago, Maritime Antarctica (Appendix A). Leaves of *Colobanthus quitensis* were collected inside the Antarctic Specially Protected Area (ASPA) 128 using permits provided by the Chilean Antarctic Institute (INACH) and by the Italian National Agency for New Technologies, Energy and Sustainable Economic Development—Technical Antarctic Unit (ENEA-UTA) during the austral summer 2020. Leaves from eight to ten individuals were collected in triplicate in three different sites:Site 1 (62°9′43.33′′ S; 58°27′58.80′′ W), about 90 m from the coast. It is strongly influenced by fauna such as penguins, seals, and elephant seals which deposit guano and feces making the soil richer in nutrients. Furthermore, the proximity of the sea affects the soil salinity.Site 2 (62°9′49.15′′ S; 58°28′9.60′′ W), 300 m from the coast and 20 m above sea level; the soil of this site is well drained.Site 3 (62°9′52.90′′ S; 58°28′21.31′′ W), located about 550 m from the coast and 30 m above sea level. This site is very windy and scarcely covered by vegetation dominated by lichens and isolated individuals of *C. quitensis* and *D. antarctica*; the ground is stony and rocky.

Freshly collected leaves were weighed and soaked in RNAlater^®^ solution (Sigma-Aldrich, St. Louis, MO, USA) in a ratio of 2:10, *w*/*v*, as indicated by the manufacturer, to stabilize and protect RNA and proteins from degradation. The samples were frozen and transported to Italy following the cold chain. Upon arrival, the samples were thawed, drained from the RNAlater^®^ solution, finely pounded under continuous addition of liquid nitrogen, and stored at −80 °C until use.

### 3.2. Protein Sample Preparation

One gram of leaf powder from each site was suspended in a lysis buffer containing 10% TCA in acetone and 10 mM DTT, left for 2 h at −20 °C, and then centrifuged at 13,500 rpm for 20 min at 4 °C. Pellets were washed in acetone, containing 10 mM DTT, 2 mM EDTA, and 1 mM PMSF, and centrifuged again under the same conditions. The obtained pellet was dried in a Speed Vac Concentrator (Savant, ThermoFisher Scientific, Waltham, MA, USA). Samples were solubilized in 100 mM Tris-HCl, pH 8.5, containing 8 M urea and 7.5 mM DTT. Sample solubilization started with a first vortex step (5 min), and then they were sonicated by using 2 min cycles (6 times) at 40 KHz and 4 °C. The samples were centrifuged at 15,000 g for 40 min at 4 °C, and then the supernatant was recovered and centrifuged again at 15,000 g for 20 min at 4 °C. Protein quantification was conducted by BCA assay (Thermo Scientific, Rockford, IL, USA) in triplicate. Disulfide bridge reduction was performed by 10 mM DTT for 45 min at 30 °C. Alkylation was obtained by 50 mM 2-iodoacetamide for 20 min, under the dark. The buffer exchange was performed using Amicon Ultra-0.5 centrifugal filters (Merck Millipore Ctd, Ireland) using an ammonium bicarbonate 50 mM buffer solution pH 7.4. 

Protein digestion was performed by treating the diluted samples with trypsin (Promega, Fitchburg, WI, USA) in an enzyme-to-protein ratio of 1:50, incubating the samples overnight at 37 °C. Reactions were stopped by adding TFA to a final concentration of 0.5%. The resulting digestion mixtures were dried in a Speed Vac Concentrator (Savant). 

### 3.3. Mass Spectrometry

Peptide digests were resuspended in loading buffer (3% acetonitrile in 0.1% formic acid). In addition, all replicates per each condition were mixed into a single pool for subsequent spectral library generation. The iRT kit (Biognosys AG, Schlieren (Zurich), Switzerland) was added to all samples according to the manufacturer’s instructions. In total, 2 µL peptide mixtures were loaded onto a 40 cm x 75 µm i.d. analytical column packed in-house with 1.9 µm C18 beads (Dr. Maisch, Reprosil-Pur 120) and separated using a linear gradient ranging from 3 to 30% buffer B (95% ACN in 0.1% FA). The flow rate is set to 300 nL/min throughout the gradient. An EASY-nLC 1200 system (ThermoFisher Scientific, Waltham, Massachusetts, US) was used, and the column was heated to 50 °C using an integrated PRSO-V1 column oven (Sonation, Biberach an der Riß, Germany) interfaced online with an Orbitrap Eclipse Tribrid mass spectrometer. For data-dependent acquisition (DDA) measurements of pooled samples, the mass spectrometer was operated in positive ion mode with an electrospray voltage of 2500 V. The survey MS1 scans were acquired over a mass range of 350–1400 m/z with an Orbitrap resolution of 120,000 using a normalized automatic gain control (AGC) target of 200 % (maximum injection time: 100 ms). DDA with a cycle time of 3 seconds was used to generate MS2 spectra using a 30 % HCD collision energy at an Orbitrap resolution of 30,000. All multiply charged ions (charge states 2–7) were accumulated with a normalized AGC target of 200 % for up to 54 ms and with a dynamic exclusion for 60 s. For data-independent acquisition (DIA) runs, 41 variable-width DIA isolation windows with a 1 m/z overlap between windows (see Table 1) were defined. DIA-MS2 spectra were acquired at an Orbitrap resolution of 30,000 over a scan range of 150–2000 m/z and a normalized AGC target of 400 % for each window. The maximum injection time was set to 54 ms. The acquisition of MS1 scans was carried out using the same settings as described for DDA-MS.

All DDA and DIA files were used to generate a spectral library using Spectronaut 14 (Biognosys AG, Schlieren (Zurich), Switzerland) against a reference FASTA file (in-house *Colobanthus quitensis* protein database [23]) with the default settings. In brief, Trypsin/P was set as a digestion enzyme with up to 2 missed cleavages allowed. Carbamidomethylation of cysteine residues was set as a fixed modification, and oxidation of methionine and acetylation (protein N-terminus) were set as variable modifications. DIA runs were searched against the generated spectral library in Spectronaut using the default settings, but protein quantification was performed on the MS2 level using only tryptic peptides and no data imputation. Data analysis was performed using custom scripts in R (64-bit version 4.1.0, Bell Laboratories, Lucent Technologies, New Jersey, created by J. Chamber) with a package *protti* (0.2.1). In brief, median-normalized protein intensities were log_2_-transformed, and statistical hypothesis testing was performed with a moderated t-test based on the R package *limma* (version 3.16, Bioconductor, created by M. Morgan). Proteins that fulfilled the following cutoffs (*p*-value < 0.05 and absolute log_2_ fold-change >0.58) were considered significantly regulated. The mass spectrometry proteomics data were deposited to the ProteomeXchange Consortium via the PRIDE [84] partner repository with the data set identifier PXD037324.

### 3.4. Bioinformatic Analysis

The differentially expressed proteins obtained from the proteomic analysis were categorized by the Gene Ontology (GO) enrichment analysis conducted using the freely available ShinyGO v. 0.76 (http://bioinformatics.sdstate.edu/go/, accessed on 26 April 2022) tool [26]. The Arabidopsis Information Resource (TAIR) code of the Arabidopsis thaliana orthologous proteins was used as input in the ShinyGO tool using the *A. thaliana* genome assembly as background. Statistical analysis included Fisher’s test and the Yekutieli multiple-test with a threshold of FDR = 0.05. Protein–protein interaction (PPI) networks were analyzed using the freely available tool kit STRING, version 11.5, (http://string-db.org/, accessed on 15 July 2022) [85]. Networks were performed at a 0.7 confidence level. Mapping of the differentially expressed proteins and pathway analysis were carried out through the Kyoto Encyclopedia of Genes and Genomes (KEGG) pathway database (http://www.kegg.jp/kegg/pathway.html, accessed on 20 June 2022).

### 3.5. Enzymatic Activity Assays

One gram of *C. quitensis* leaves grown in the three sampling sites was used for protein extraction, employing three biological replicates. The leaves were finely ground using a mortar and pestle under continuous addition of liquid nitrogen, and then 5 mL of the following protein extraction buffer was added: 50 mM cold sodium phosphate buffer (pH 7.6) containing 1 mM EDTA, 4% (*w*/*v*) polyvinylpyrrolidone, 3 mM DTT, and a cocktail of protease inhibitors (Complete ULTRA tablets, Roche). The homogenate was centrifuged (Universal 32R Hettich) at 9000 rpm for 15 min at 4 °C, and the supernatant was used for enzyme activity assays as already described [86]. All enzymatic activities were performed on three biological replicates using also three technical replicates for each sample to assess the error of the technique. The protein content was estimated according to the method described by Bradford [87] using BSA as a standard. 

Catalase (CAT) activity was measured using the method of [88]. The reaction mixture (1 mL final volume) contained a 50 mM potassium phosphate buffer (pH 7.6) and 19 mM H_2_O_2_; the reaction was started by adding 30 µL of protein extract. The decrease in absorbance due to the decomposition of H_2_O_2_ (ε = 0.0436 mM^–1^ cm^−1^) was monitored at 240 nm. The activity of glutathione-S transferase (GST) was determined by measuring the absorption at 340 nm of reduced glutathione (GSH; 1 mM) conjugated with 1-chloro-2,4-dinitrobenzene (CDNB; 1 mM) (ε = 9.6 mM^−1^ cm^−1^) in 100 mM potassium phosphate buffer (pH 6.5) containing 1 mM EDTA. The activity of guaiacol peroxidase (POD) was measured at 470 nm following the formation of tetraguaiacol (ε = 26.6 mM^−1^ cm^–1^) due to guaiacol reduction. The reaction buffer contained guaiacol (0.4% *v*/*v*) and H_2_O_2_ (0.03% *v*/*v*) in 100 mM potassium phosphate buffer (pH 7.6). All the above enzyme activities were expressed as Unit mg^−1^ protein. To measure the activity of superoxide dismutase (SOD), the kit from Sigma (Sigma-Aldrich, Uppsala, Sweden) was used which is based on the use of the tetrazolium salt WST-1 (2- (4-iodophenyl) -3- (4-nitrophenyl) -5- (2,4-disulfophenyl) -2H-tetrazolium) which produces a water-soluble formazan dye detectable at 440 nm upon reduction with a superoxide anion. The rate of reduction of WST-1 by the superoxide anion is linearly related to the activity of xanthine oxidase and is inhibited by SOD. Hence, the SOD activity can be determined by measuring the decrease in formazan which is proportional to the SOD inhibiting activity. Therefore, the inhibition activity corresponds to the quantity (µg) of total protein extract needed to reduce the formation of formazan by 50% (IC50). Low IC50 levels are correlated with high concentrations of SOD.

### 3.6. Lipid Peroxidation Damage

To evaluate the damage due to lipid peroxidation, the level of reactive substances of thiobarbituric acid (TBARS) was measured [89]. Using a mortar and pestle, about 400 mg of frozen leaves was finely ground by continuously adding liquid nitrogen. The powder was resuspended in 3 mL of 0.1% trichloroacetic acid (TCA), mixed on the vortex until complete homogenization, and centrifuged at 13,500 rpm for 10 min. Four hundred microliters of the supernatant (or 0.1% TCA for blank) was added to 1 mL of 0.5% TBA in 20% TCA (+TBA solution) or to 1 mL of 20% TCA (-TBA solution) (factor dilution 1: 3.5). After 30 min incubation at 80 °C, the samples were cooled on ice and then centrifuged at 13,500 rpm for 5 min. The absorbance of the TBA-TBARS complex was measured at 532 nm while the absorbance at 600 nm was measured to allow correction of non-specific turbidity. The ε_µM_ of malondialdehyde (MDA), one of the main products of membrane damage, was used to calculate TBARS equivalents (nmol mL^−1^) according to the following formula:(1)$$\frac{A}{\varepsilon_{\mu M}\;\mathrm{MDA}}\;\times \;\mathrm{DF}$$
where the dilution factor (DF) is 3.5, ε_µM_ MDA is 0.155 µM^−1^ cm^−1^, and A is the absorbance calculated through the following equation:[(Abs532_+TBA_) − (Abs600_+TBA_)] − [(Abs532_−TBA_) − (Abs600_−TBA_)]
where Abs532_+TBA_ is the absorbance at 532 nm for the +TBA solution, Abs600_+TBA_ is the absorbance at 600 nm for the +TBA solution, Abs532_−TBA_ is the absorbance at 532 nm for the −TBA solution, and Abs600_−TBA_ is the absorbance at 600 nm of the −TBA solution.

### 3.7. RNA Extraction and Quantitative Reverse Transcriptase-PCR Analysis

The Nucleospin^®^ RNAPlant kit (Macherey-Nagel, Düren, Germany) was used to extract total RNA from 100 mg of finely pounded leaves. RNA concentration was estimated by reading spectrophotometric absorbance at 260 nm (spectrophotometer UV-30 SCAN, ONDA). The OD_260_/OD_280_ nm and OD_260_/OD_230_ nm absorption ratios were calculated to evaluate RNA quality and purity whose integrity was also verified by agarose gel electrophoresis. The absence of DNA contamination was tested by semi-quantitative PCR using 100 ng of total RNA as a template and EF1α specific primers for amplification. The ImProm-II™ Reverse Transcription System (Promega, Madison, WI, USA) was used to synthesize cDNA starting from 1 µg of RNA as a template and using the oligo-dT primer for first strand synthesis. To obtain the sequences of the genes to be amplified, the Shotgun Transcriptome Assembly of *C. quitensis* leaves, performed by our research group [22] (National Center of Biotechnology Information (NCBI), Sequence Read Archive (SRA), accession N. SRX814890) was used. Primer3 software (http://bioinfo.ut.ee/primer3-0.4.0/, accessed on 16 May 2022) was used for primer pair design. The cDNAs were synthesized using RNA as a template, and the BIOTAQ DNA polymerase (Bioline, London, UK) was used to verify the specificity of each primer pair by standard PCR. All primers used in the present study are listed in Table 2.

A Bio-Rad CFX96 Real-Time PCR thermal cycler (Bio-Rad, Hercules, CA, USA) was used for quantitative reverse transcriptase-PCR (qRT-PCR) reactions, performed in 96-well plates using the SYBR Green detection system. The reaction mixture (10 µL) contained 1 µL of four-fold diluted cDNA, 5 µL Sso Advanced SYBR Green Supermix (Bio-Rad, Hercules, CA, USA), and different concentrations of each gene-specific primer. The following cycling conditions were used: initial denaturation step at 95 °C for 3 min, followed by 40 cycles at 95 °C for 10 s and primer-specific annealing temperature for 30 s. The melting curves (ranging from 70 to 95 °C with a steady increase of 0.5 °C every 5 s) were analyzed to examine the PCR specificity. Gene expression of selected genes was normalized against the reference genes elongation factor 1-alpha (*EF1α*), as suggested in [90]. Quantitative analysis was performed according to the 2^−∆∆Cq^ method. Real-time DNA amplifications were processed using the CFX ManagerTM Software (Bio-Rad, Hercules, CA, USA). Results of qPCR analyses were verified in three independent biological experiments using three technical replicates each.

### 3.8. Statistical Analysis

The statistical significance of data obtained from enzymatic activity assays and gene expression analysis was assessed using one-way ANOVA and Tukey’s multiple comparison test. Three biological replicates were used, expressing the values as means ± standard deviation (SD). The significance threshold was set at *p* <0.05.

## 4. Conclusions

In conclusion, our results show that *C. quitensis* finely regulates its metabolic activity and stress response capacity as a function of the environment. The main results are shown in Figure 8. 

In summary, our results show that Site 2 plants have a highly active energy metabolism that likely supports growth and stress tolerance. Furthermore, Site 2 appears to be the most suitable to deal with oxidative stress by activating ROS-scavenger enzymes that, in turn, limit lipid peroxidation. Furthermore, they are less prone to ER stress and protein denaturation/misfolding and appear to better counteract biotic stresses by activating the immune response. Conversely, general growth-related metabolic pathways appear to be underrepresented in Site 3 plants facing a more hostile environment. Furthermore, failing to counteract oxidative and ER stresses, these plants could be less well protected from potential stressors of both biotic and abiotic nature and, consequently, more damaged. On the other hand, plants growing in Site 1 need to face salinity and osmotic stress and, like Site 3 plants, also appear to suffer from ER stress and protein denaturation/misfolding. Nevertheless, these plants are able to activate the immune response, although, unlike plants growing in Site 2, they do not undergo intense metabolic activity.

The results obtained during this work are very interesting and pave the way for new insights into the influence of the environment on plant fitness and tolerance to stress. In the future, it would be interesting to deepen knowledge on the characteristics of the soils (chemical components and microbiota) which certainly greatly impact plant growth and wellness.

## Figures and Tables

**Figure 1 plants-11-03176-f001:**
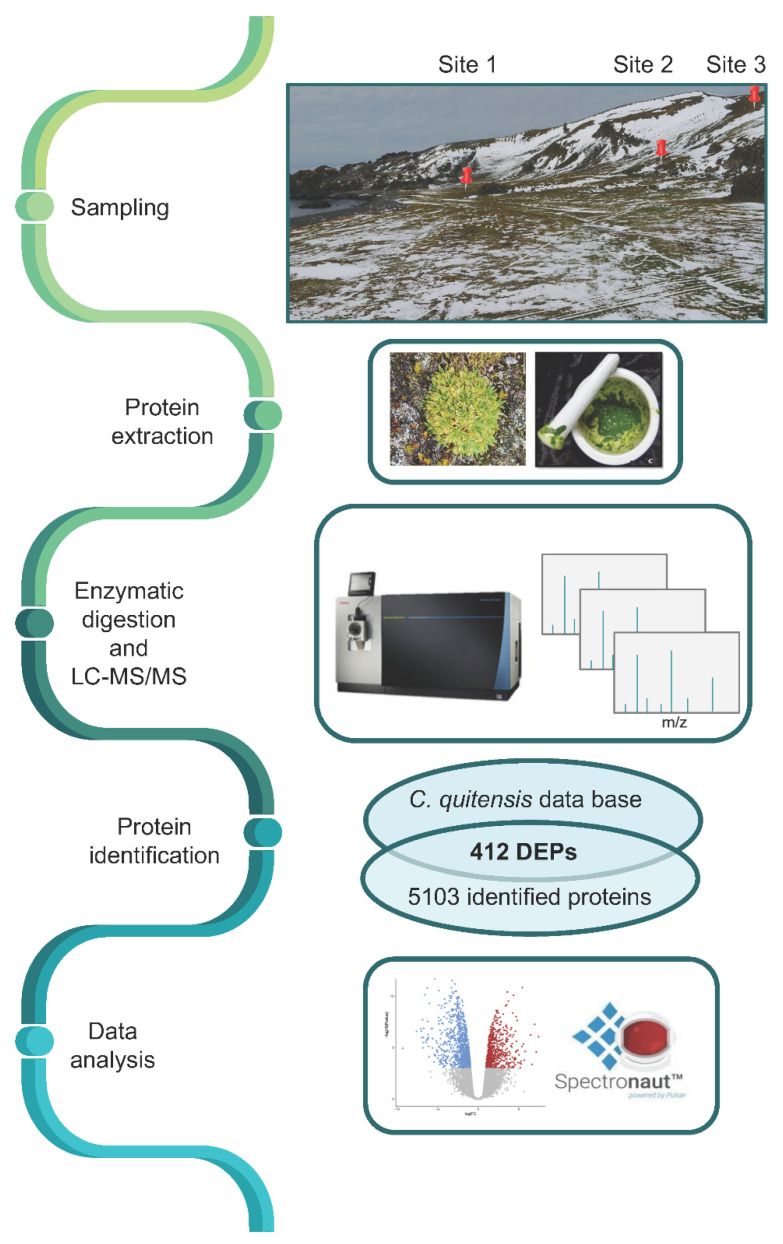
Experimental workflow (left); pictures on the right refer to each step of the workflow.

**Figure 2 plants-11-03176-f002:**
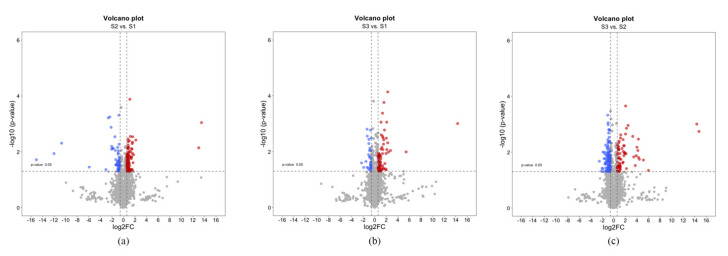
Volcano plot representation of differentially expressed proteins (DEPs) in the three data sets. Statistically significant (*p*-value < 0.05), |log2FC| > 0.58) upregulated and downregulated proteins for each data set are indicated with red and blue dots, respectively. Non-significant proteins are represented in gray. (**a**) S2 vs. S1 data set; (**b**) S3 vs. S1 data set; (**c**) S3 vs. S2 data set.

**Figure 3 plants-11-03176-f003:**
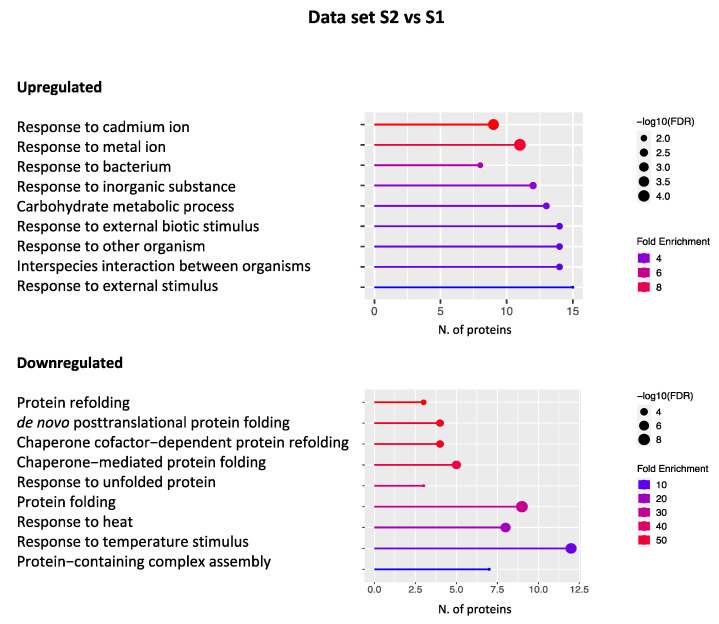
Graphical representation of the GO terms enriched in the S2 vs. S1 data set. The *x*-axis refers to the number of DEPs upregulated or downregulated that significantly enrich a specific GO term. Lines are colored by a red–blue gradient based on fold enrichment, whereas the size of the circle at the end of the lines is related to the statistical significance (−log10 FDR).

**Figure 4 plants-11-03176-f004:**
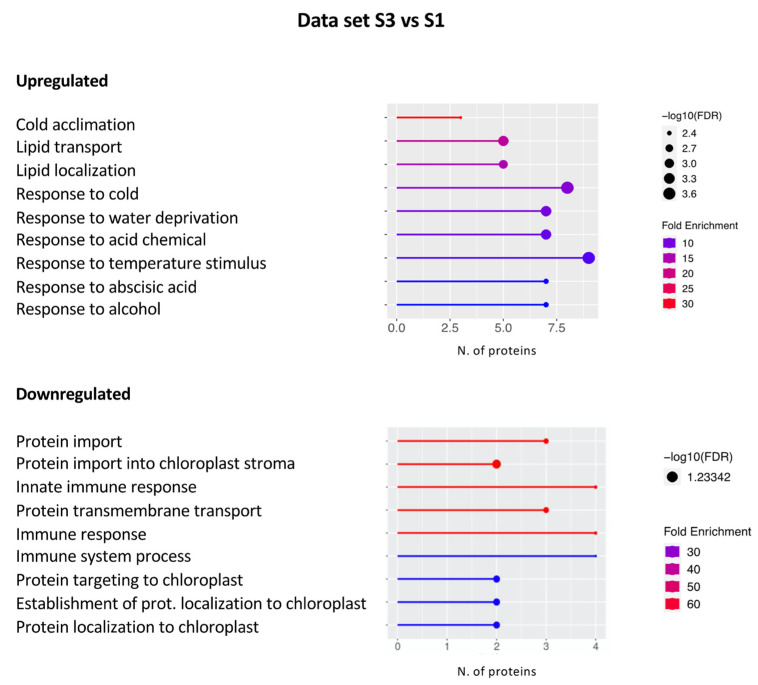
Graphical representation of the GO terms enriched in the S3 vs. S1 data set. The *x*-axis refers to the number of DEPs upregulated or downregulated that significantly enrich a specific GO term. Lines are colored by a red–blue gradient based on fold enrichment, whereas the size of the circle at the end of the lines is related to the statistical significance (−log10 FDR).

**Figure 5 plants-11-03176-f005:**
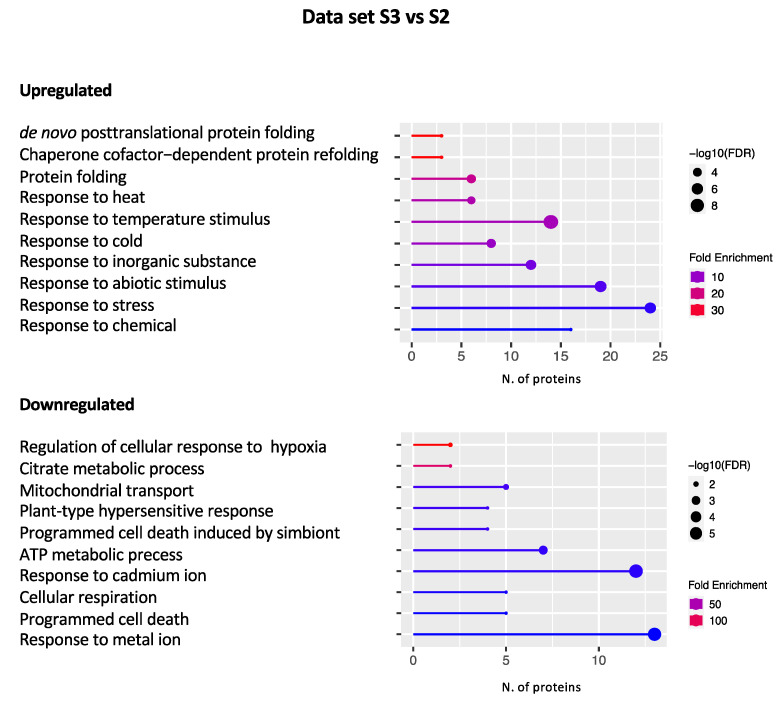
Graphical representation of the GO terms enriched in the S3 vs. S2 data set. The *x*-axis refers to the number of DEPs upregulated or downregulated that significantly enrich a specific GO term. Lines are colored by a red–blue gradient based on fold enrichment, whereas the size of the circle at the end of the lines is related to the statistical significance (−log10 FDR).

**Figure 6 plants-11-03176-f006:**
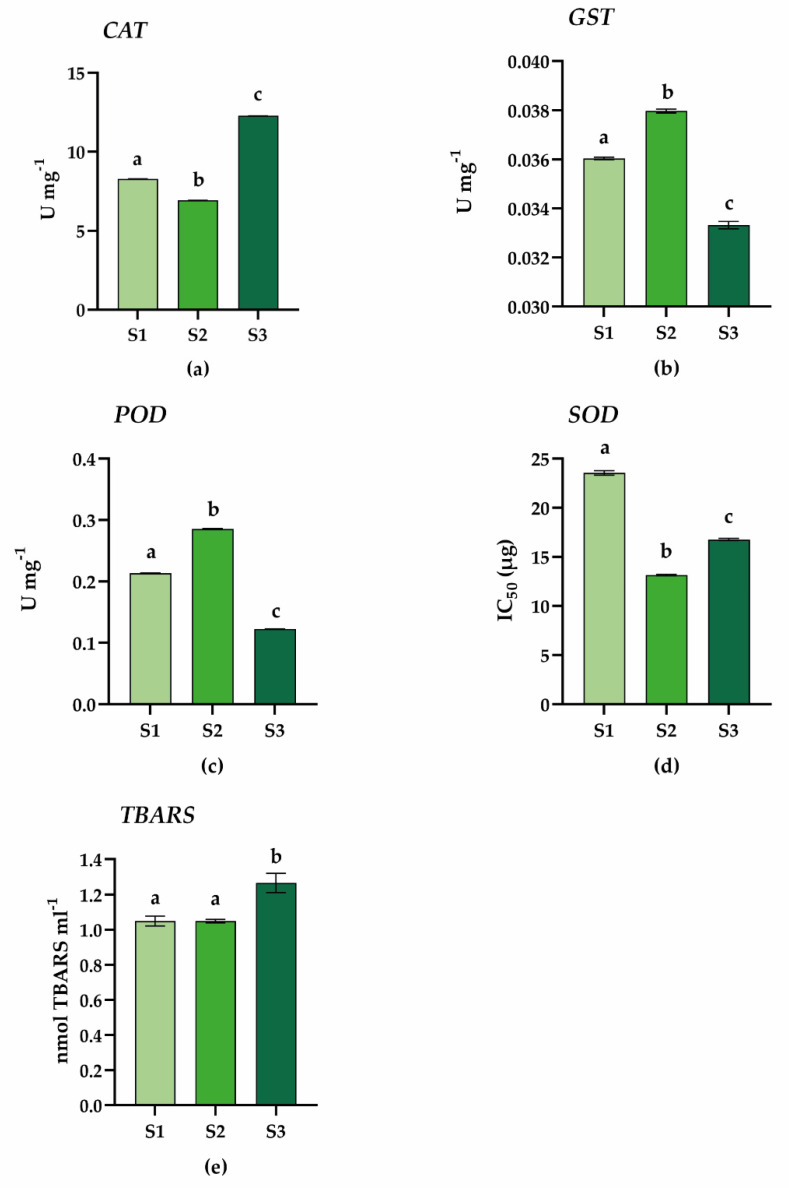
Antioxidant enzyme activities and TBARS content in *C. quitensis* leaves collected at the three sites S1, S2, and S3 (x-axes). (**a**) Catalase (CAT); (**b**) glutathione S-transferase (GST); (**c**) guaiacol peroxidase (POD); (**d**) superoxide dismutase (SOD); (**e**) TBARS content. Data represent the mean ± SD of three biological replicates. For all enzymatic assays, a statistically significant difference was found between all sites (*p* < 0.0001). As for TBARS, a statistically significant difference was found only between S1 vs. S3 and S2 vs. S3 (*p* < 0.0001). Statistic test: one-way ANOVA, Tukey’s multiple comparisons. The letters indicate significant differences between the samples.

**Figure 7 plants-11-03176-f007:**
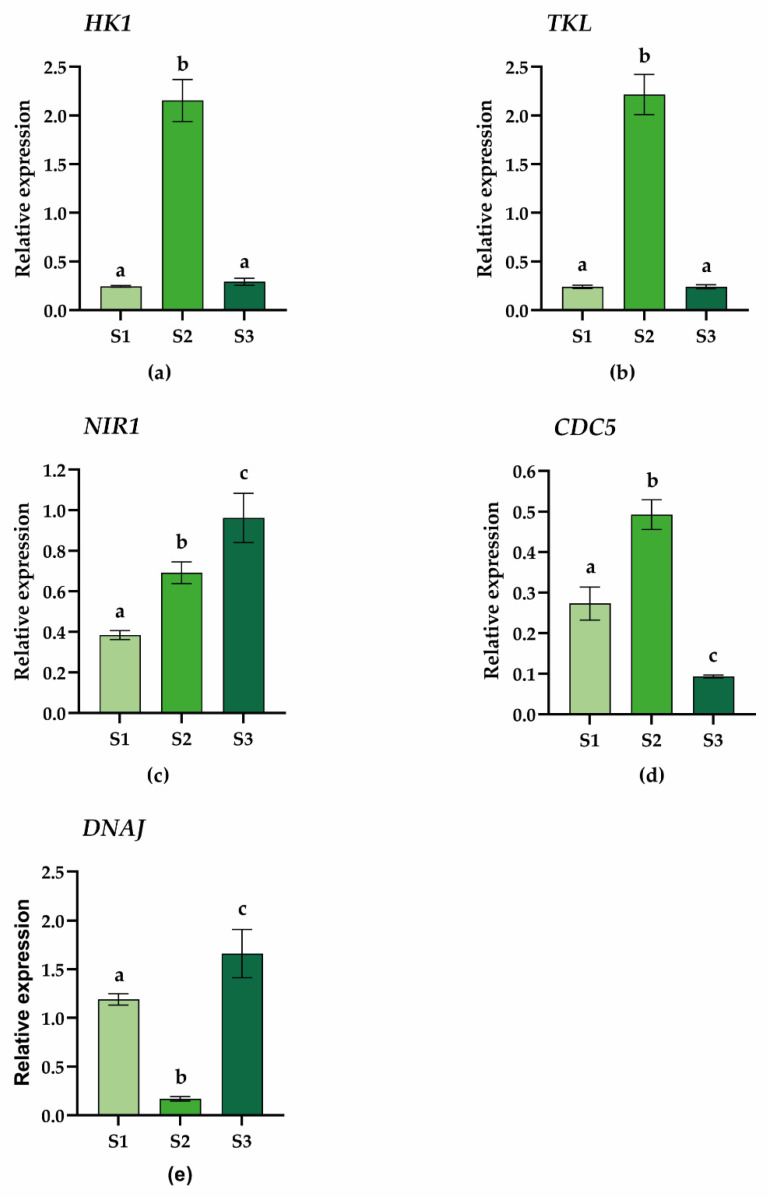
Relative expression of *C. quitensis* genes normalized with the reference genes elongation factor 1-alpha (*EF1α*). (**a**) Hexokinase (*HK1*); (**b**) transketolase (*TKL*); (**c**) nitrite reductase 1 (*NIR1*); (**d**) cell division cycle 5 (*CDC5*); (**e**) *DNAJ*. Error bars represent the mean ± SD of three biological replicates. Statistic test: one-way ANOVA, Tukey’s multiple comparisons (*p* < 0.05). The letters indicate significant differences between the samples.

**Figure 8 plants-11-03176-f008:**
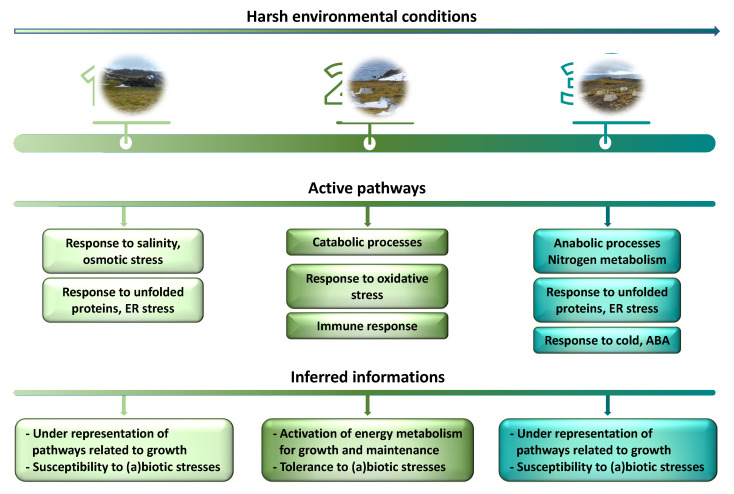
Outline of the main results obtained in the present paper.

**Table 1 plants-11-03176-t001:** Characteristics of the three sampling sites.

Parameters	Site 1	Site 2	Site 3
Mean Air Temperature (°C)	2.7 ± 0.1	2.1 ± 0.1	2.1 ± 0.1
Mean Max Air T (°C)	5.3 ± 0.4	4.9 ± 0.4	5.1 ± 0.4
Mean Min Air T (°C)	0.6 ± 0.3	0 ± 0.3	0 ± 0.3
Ground Temperature	2–4 °C higher than air T		
Wind (m/s)	5	7	10
Distance from the Coastline (m)	90	300	550
Altitude (masl)	0	20	30
**Soil composition %**			
Sand	73	83	57
Silt	19	11	27
Clay	9	6	16
Textural Class	Sandy loam	Loamy sand	Sandy loam
pH	4.77	5.37	6.11
Classification	Very strongly acidic	Strongly acidic	Slightly acidic

**Table 2 plants-11-03176-t002:** List of primers for confirmatory analyses by qRT-PCR.

Gene Target	Acronym	Primers	Annealing Temperature (°C)	Amplicon Size (bp)
Heat shock protein 40	DNAJ	For: GCCTCAACAAGTGATGCTTTC	50	103
		Rev: CTCCAGCCGACTTAGTCTTTATT		
Cell division cycle 5	CDC5	For: ACTTGAGAGACCATAGGCATTAC	50	102
		Rev: GGACATGAAGAGGACTCACTTG		
Nitrite reductase 1	NIR1	For: CCGTCACAAACTGCGAAATAAG	54	118
		Rev: CTTCAGAGTGGCATGGACAA		
Hexokinase 1	HK1	For: GCTTCTGAAGGCGGTTCTAA	50	100
		Rev: CACCAAGGTCCAAAGCATAGA		
Transketolase	TKL	For: GACCCAGCTTCGATGCTAAC	50	151
		Rev: CCCAAGCAGGTGATGAACTT		

## Data Availability

The mass spectrometry proteomics data were deposited to the ProteomeXchange Consortium via the PRIDE partner repository with the data set identifier PXD037324.

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
