# Peer review of "Environmental Signals Act as a Driving Force for Metabolic and Defense Responses in the Antarctic Plant *Colobanthus quitensis"

_plants, 2022, doi:10.3390/plants11223176_

Round 1
Reviewer 1 Report
The manuscript of Bertini et al., investigated the defense responses in the Antarctic plant Colobanthus quitensis based on environmental signals that acts as a metabolic and defense response. The effect of environmental stimuli on plant growth and defense mechanisms is highly relevant, especially of plants located in very remote areas, Antarctica in this case. Therefore this paper deserves recognition in this field of research. The approach of the authors is well performed, the article is well written and the applied methodology merits publication in Plants. However, after reading the manuscript I have minor comments and 1 major comment which are ought to be addressed before publication:
- Please mind consistency throughout the text. Examples: (i) use a space between number and unit, (ii) scientific names of plants in italic, (iii) sometimes a specific amount is presented as a number, sometimes as text, (iv) sometimes a chemical element is presented as a symbol, sometimes as text, (v) use subscripts when presenting chemical formulas. Examples (but corrections should not be limited to these examples, please go through the text): Lines: 40, 41, 75, 83, 85, 128, 203, 205, 206, 210, 295, 311, 362, 105, 385, 537, 518, 565, 648.
- Something went wrong with Figure 2, please correct.
- What is meant with biological and technical replicates? Is the technical replicate performed to assess the error of the analytical measurement technique? Please elaborate. Which specific SD is depicted on Figure 6 and 7 ?
- How is the soil composition of the 3 sites determined ? Please elaborate.
- Did the authors consider ICP analysis (of the soil), since iron and cadmium responses are specifically mentioned ?
- Which power output was applied for the sonication of the sample solubilization? (section 3.2).
- Please have a look at Equation 1. It is not clear how a unit of nmol mL-1 is obtained by applying Eq. 1.
- What is meant with the 2-∆∆Cq method? (Line 704).
- In the conclusion section Lines 715-722 are redundant as the information is already provided elsewhere in the manuscript.
Reviewer 2 Report
The manuscript “Environmental signals act as a driving force for metabolic and defense responses in the Antarctic plant Colobanthus quitensis is dealing with metabolomic analysis of plants lived in Antarctic region in three different sites with progressive environmental change. By using a combination of high-throughput proteomic profiling techniques and bioinformatics tools 412 differentially expressed proteins have been identified. Authors showed different modulation of metabolic, oxidative and defense stress responses between different environmental sites. The manuscript has significant scientific merit and falls in a scope of the journal, but it is not acceptable for the publication in a present form.
Line 157-160: The main problem is Figure 2. What is presented in Figure 2? The description in Figure legend does not correspond to presented plot. It looks like that presented plot is incomplete. This is one of the most important results but data are missing. Please provide complete results about upregulated and downregulated proteins for each data set (a) S2 vs. S1 data set; (b) S3 vs. S1 data set; (c) S3 vs. S2 data set indicated with red and blue dots significant proteins while non-significant proteins should be presented in grey.
Lines 77, 83: Latin names of species in italic.
Line 107-121: This part is more suitable for Introduction section
Round 2
Reviewer 1 Report
The authors revised the manuscript accordingly and therefore the reviewer accepts this manuscript for publication in Plants.
Author Response
Thank you very much for your time and competence
Reviewer 2 Report
The authors significantly improved first version of manuscript.
Author Response

(The authors gave the same response as above.)
